# Gender Influences on Physical Activity Awareness of Adolescents and Their Parents

**DOI:** 10.3390/ijerph18115707

**Published:** 2021-05-26

**Authors:** Joaquín Lago-Ballesteros, Marcos García-Pascual, Miguel Ángel González-Valeiro, María Á. Fernández-Villarino

**Affiliations:** 1Department of Applied Didactics, Faculty of Teacher Training, Universidade de Santiago de Compostela, 27002 Lugo, Spain; joaquin.lago@usc.es; 2Department of Special Didactics, Faculty of Education and Sports Sciences, Universidade de Vigo, 36004 Pontevedra, Spain; marcosgpascual@gmail.com; 3Departent of Physical and Sports Education, Faculty of Sports Sciences and Physical Education, Universidade da Coruña, 15179 A Coruña, Spain; maglez@udc.es

**Keywords:** gender differences, physical activity, assessment imbalances, adolescents

## Abstract

The imbalances between the actual physical activity (PA) of adolescents and the subjective perception both they and their parents have in this regard can play an important role in perpetuating inactive lifestyles. The aim of this study is to analyse these discrepancies by considering gender as a conditioning factor. The participants in the study were 1697 adolescents, 1244 mothers and 1052 fathers in the educational communities of 26 secondary schools located in urban environments of the Autonomous Community of Galicia (Spain). With regard to actual physical activity, a high prevalence of sedentarism (82.1%) was revealed, this being even more acute in girls (87.8%). However, the perceived levels of activity differed significantly from the actual ones with a clear general overestimation both by the adolescents and their parents. When further exploring the data, gender influences were also detected both in adolescent and parental perceptions, since the high rates of overestimation in sedentary individuals were lower in girls and, on the contrary, the low rates of underestimation in active individuals were higher in girls. Moreover, although the level of agreement between actual and perceived physical activity was low overall, with Cohen’s kappa values ranging from 0.006 to 0.047, the lowest values were observed in the case of girls. In conclusion, both the adolescents and their parents were incapable of correctly assessing the actual physical activity of the former, so it seems that the general population lacks knowledge about the amount of physical activity that adolescents need to do to achieve a healthy lifestyle. Consequently, it would be advisable to implement health education campaigns and awareness-raising interventions directed to young people as well as to their parents and, in doing so, gender must be considered by establishing distinct program designs in terms of this variable.

## 1. Introduction

In 2010, the WHO signalled that, on a global scale, 81% of adolescents between the ages of 11 and 17 were sedentary [1]. Subsequent studies describe only a slight evolution in the data [2]. Considering gender, the situation is particularly bad for girls, who increased their prevalence of physical inactivity, even setting it at over 83% [3].

The factors that contribute to the maintenance of these figures are numerous and have been studied by different institutions and researchers [4]. As a part of this effort, a new line of research emerged to address subjective perception of physical activity (PA) as a factor that can significantly contribute to the reproduction of behavioural models that steer adolescents away from active lifestyles. As the basic concept of this body of research, perception refers to sensory awareness and depends on the knowledge and opinions held by the individual about what it is being assessed [5,6,7]. If we talk specifically about perception of PA, subjectivity is also present [8]. That is, when people are asked to assess their own level of PA, their previous knowledge could determine the realism of their assessments [9]. In this way, two people who demonstrate similar PA habits may differ in their assessments in accordance with previously acquired knowledge [10].

A part of previous literature has considered subjective perception of PA from the perspective of adolescents themselves, and it has been shown that a high subjective attribution of importance to PA could be a key element to achieving and maintaining an active lifestyle [11]. To this regard, Greca et al. [12] pointed out that it is important for these young individuals to perceive themselves as active on their own merits when doing PA, with the aim of eliminating the rate of sedentarism. For their part, Van Sluijs, Griffin and Van Poppel [13] affirm that people who consider themselves to be more active than they really are have a healthier lifestyle than those who perceive themselves as sedentary. Only a few studies have analysed the PA awareness of European adolescents [14,15]. According to their findings, 60.3% to 80.9% of adolescents believe that they are physically active when they are not. When considering possible gender differences, it has been observed in some studies that boys perceive themselves to be up to 3.8 times more active than girls [16]. Moreover, it has been pointed out that in the 12–17 age group, girls with higher levels of body dissatisfaction and worse BMIs (Body Mass Index) reveal a more devalued physical self-concept [17], which could lead them to do less PA [18,19,20,21]. Notwithstanding, previous studies have scarcely considered potential gender influences over the realism of adolescents’ assessments. To this regard, Corder and colleagues [14] found that, compared with girls, boys were less likely to overestimate their PA.

In another brand of research, subjective perception of adolescents’ PA was considered from the perspective of their parents and, to this regard, Wash et al. [22] highlighted the potentially important role of parents’ perceptions in tackling their children’s sedentary habits. Other authors consider that when parents portray their offspring as being active when in fact they are not, then their physical habits will never change [23]. But parents do not only have a potentially negative effect since they can also represent a positive influence that helps their children to achieve an active lifestyle [10], characterised by including a daily minimum of 60 min of moderate-to-vigorous intensity PA, mainly aerobic, along with muscle-strengthening activities carried out two to three times a week [24]. However, it has been described that parents commonly overestimate the amount of PA their children do [25] and, where this occurs, the perception of parents could contribute towards perpetuating the problem of inactivity [26,27]. When analysing whether the parental influence is dependent on their gender, some studies suggest that the role of fathers with regard to the influence on their children’s PA habits takes precedence over that of mothers, and it is also more effective in promoting a healthy lifestyle for sons as opposed to daughters [28,29,30]. Nevertheless, the differences in parents’ perceptions that might result from the gender of their offspring remain unexplored. Besides, to the authors’ knowledge, no previous studies have examined the level of agreement between PA assessments by adolescents and their parents.

Consequently, the aims of the present study were to: (1) Examine the PA awareness of adolescents and their parents; (2) determine the level of agreement between the adolescents’ and parental assessments; and (3) identify gender influences on the PA awareness of adolescents and their parents.

## 2. Method

### 2.1. Participants

A purposive non-probability sample was used to pick the participants, whereby 26 schools located in urban settings in the Autonomous Community of Galicia (Spain) were chosen. Schools were selected for their belonging to the *Sports Project for the Centre* program, a regional government-sponsored initiative to promote active lifestyles that aims to ensure an adequate provision of PA in schools. In order to maximise the possibility of obtaining a sample that would cover the different socio-economic levels present in the target population, it was verified that schools were located in different neighbourhoods of their urban areas.

Once the list of schools of interest was established and permission was granted from the principals, both pupils and their respective parents were given the opportunity to participate anonymously and voluntarily. The sample finally comprised 1697 adolescents (873 girls, that is, 51%) and 2296 parents (1244 mothers, that is, 54%). The pupils’ ages ranged from 12–18 (14 ± 1.1 for the boys and 14 ± 1.7 for the girls). Table 1 shows the socio-demographic characterisation of the parents.

### 2.2. Variables

Table 2 shows the dependent variables considered, while the only independent variable was gender (male, female).

### 2.3. Data Collection Instrument

The International Students’ Lifestyle Questionnaire (LSQ), previously used in numerous studies [31,32,33,34], was used as the data collection instrument. This questionnaire comprises 39 closed-ended questions, structured in 4 dimensions: (1) personal data (6 items), (2) lifestyle habits (12 items), (3) attitudes and perceptions (12 items), and (4) evaluation of school, physical education and PA (9 items). The analysis included in this study focuses on dimensions 2 and 3.

Cronbach’s Alpha was used as an internal consistency index to evaluate the reliability of the instrument, with values of α = 0.89 and α = 0.75 for dimensions 2 and 3, respectively.

### 2.4. Procedure

Once the schools taking part in the study were chosen, their principals were contacted by telephone and asked to collaborate. After this first contact, written detailed information about the general aims of the study and its procedures was mailed to the principals in order to clarify any doubt. The anonymous and voluntary participation of both schools and their students was emphasised. Upon acceptance, written authorisation from principals and informed consent from both the adolescents and their parents were obtained. Finally, the questionnaire was administered to each class group independently, under the coordination of each individual school and in the presence of the PE teachers. 

The study was carried out in accordance with the ethical standards of Sports Sciences [35]. The protocol was approved by the Universidade da Coruña’s Teaching and Research Ethics Committee, in the framework of implementing a wider proposal within the Euro-American Physical Activity, Education and Health Network. 

### 2.5. Data Analysis

Firstly, we performed a descriptive analysis, characterising the frequency and time the adolescents, girls and boys, spent doing PA in different contexts, through frequencies and percentages. Furthermore, the data were reduced in order to obtain an overall proxy variable of the true level of PA carried out by the adolescents. In this sense, participants were classified as active or sedentary in accordance with whether or not they complied with the minimum recommendations established by the WHO for their age group [24]. In order to do this, participants were classified as active if their reports of PA (considered overall as the sum of the different types considered) comprised a minimum of 7 h and 7 times a week. Where this condition was not fulfilled, participants were classified as se-dentary. This proxy variable was also described through frequencies and percentages. The PA done by adolescents was also subject to a comparison in accordance with gender using the Chi-square test. 

We then performed a descriptive analysis of adolescents’ and parents’ perceptions regarding the PA level of the former. Frequencies and percentages were calculated for each gender group and compared via Chi-square tests. Then, Odds Ratios (ORs) were computed in order to compare the probabilities of adolescents being perceived as active according to gender. To accomplish this later analysis, previous data reduction was necessary and hence, adolescents’ and parents’ assessments were dichotomised (not active at all and scarcely active were merged as sedentary and quite active, active, and very active were combined as active).

Thirdly, we performed an analysis, both for each gender group and overall, of the level of correspondence between actual and perceived activity levels, as well as between the perceptions of adolescents and their parents and the perceptions of parents among themselves. Cohen’s kappa coefficient was used to provide a measure of agreement. 

All of the analyses described were performed using SPSS statistics for Windows version 20.0 (SPSS Inc., IBM, Armonk, NY, USA), establishing a value of *p* < 0.05 for the statistical significance of the contrasts.

## 3. Results

### 3.1. True Physical Activity Level of Adolescents

Table 3 shows the characterisation of the PA of the adolescents participating in the study. The majority of PA, for both the boys and the girls, took place outside school hours and was unsupervised. In this context, statistically significant differences were observed in terms of gender (χ^2^ = 112.961; *p* < 0.001), with a greater frequency of activity for boys, of whom 45.1% registered a frequency of four times or more per week, while the percentage corresponding to girls was only 25.8%. When analysing the activity performed under the supervision of a sports or PA professional, statistically significant differences were also found in terms of gender, both in school (χ^2^ = 100.897; *p* < 0.001) and outside school (χ^2^ = 47.311; *p* < 0.001), with a higher frequency of activity in boys. Specifically, with regard to the percentage of boys who affirmed that they did PA four or more times a week, 31.1% was in clubs or associations and 11.6% was in school but outside school hours, while in the case of the girls these percentages stood at 17.7% and 6.1%, respectively. It is also to be noted that the majority of the girls never did any PA under professional supervision (49.9% in clubs and associations and 71.4% outside school). 

Regarding the time spent doing PA, statistically significant differences were observed in terms of gender for the different scenarios considered, that is, boys spent more time than girls doing PA outside school unsupervised (χ^2^ = 94.314; *p* < 0.001), outside school supervised (χ^2^ = 21.308; *p* < 0.001) and in-school supervised (χ^2^ = 14.834; *p* < 0.01). Specifically, four or more hours a week of PA were reported by (i) 26.8% of the boys and 10.9% of the girls in the outside school unsupervised scenario, (ii) 39.5% of the boys and 22.0% of the girls in the outside school supervised scenario, and (iii) 18.9% of the boys and 9.0% of the girls in the in-school supervised scenario. The majority of the adolescents, regardless of gender and context, spent less than 2–3 h a week doing PA. 

Considering overall PA, the percentage of adolescents meeting WHO recommendations differed significantly across genders (χ^2^ = 36.511; *p* < 0.001), with that of the boys (23.7%) almost doubling that of the girls (12.2%).

### 3.2. Adolescents’ Perceived Levels of Physical Activity

Going on to analyse adolescents’ and their parents’ perceptions regarding the former’s level of PA, Table 4 represents their characterisation in terms of gender and actual level of activity. With regard to adolescents’ self-perception, statistically significant differences were found between boys and girls (χ^2^ = 116.107; *p* < 0.05), meaning that the probability of a girl considering herself active was 2.7 times lower than that of a boy (OR = 0.37, 95% CI: 0.28−0.498, *p* < 0.001). As for the parents’ perception, both the mothers (χ^2^ = 16.046; *p* < 0.05) and the fathers (χ^2^ = 23.704; *p* < 0.05) showed statistically significant differences in their perceptions of their sons’ and daughters’ PA levels. As such, the probability of a girl being considered by her mother to be active was 1.36 times lower than that of a boy (OR = 0.73, 95% CI: 0.55−0.98, *p* < 0.05), and when it was a father who made the assessment, the probability of a girl being considered active was 1.52 times lower than that of a boy (OR = 0.66, 95% CI: 0.48−0.92, *p* < 0.05).

### 3.3. Correspondence between Adolescents’ Actual and Perceived Physical Activity Levels

When analysing the level of correspondence between actual PA levels (degree of compliance with WHO recommendations) and those perceived, a poor level of concordance was observed on the part of the adolescents, both overall (*k* = 0.043) and differentiated for the boys (*k* = 0.041) and the girls (*k* = 0.031). In more detail, widespread overestimation was observed, as 90.4% of the boys and 80.4% of the girls classified as sedentary according to their actual activity perceived themselves to be active. On the other hand, the percentages of active boys and girls who underestimated their PA level, perceiving themselves to be sedentary, were as low as 1.6% and 9.1%, respectively. On the part of the mothers, the levels of concordance were also poor overall (*k* = 0.026) and specifically for the boys (*k* = 0.037) and the girls (*k* = 0.009). Although to a lesser extent than among adolescents, high percentages of overestimation were also present among mothers, who misclassified their sedentary sons and daughters as active, respectively, 82.9% and 79.8% of the time. The cases where mothers underestimated PA levels affected 10.3% and 16.6% of active boys and girls. Lastly, on the part of the fathers, the levels of concordance were poor again, overall (*k* = 0.030) and in particular for the boys (*k* = 0.047) and for the girls (*k* = 0.006). When looking at the discrepancies in detail, it was observed that fathers overestimated the PA level of their offspring in 84.5% and 81.2% of sedentary sons and daughters, while underestimating it in 6.3% and 16% of active boys and girls.

Insofar as the agreement between adolescents’ and their parents’ perceptions are concerned, poor levels were observed, both in the overall adolescent–mother (*k* = 0.296) and adolescent–father (*k* = 0.280) comparisons as well as in the specific son–mother (*k* = 0.229), son–father (*k* = 0.220), daughter–mother (*k* = 0.326) and daughter–father (*k* = 0.301) comparisons.

Lastly, when establishing the level of correspondence between the parents’ perceptions, a good concordance was found overall (*k* = 0.619), as well as separately with regard to sons (*k* = 0.609) and daughters (*k* = 0.623).

## 4. Discussion

Considering gender as a conditioning factor, the aim of this study was to analyse the level of correspondence between the actual PA level of a group of adolescents and the perceptions held by adolescents themselves as well as their parents concerning said levels. This analysis is substantiated by previous literature indicating that the discrepancies between the reality and the subjective perception of adolescents [12,13,22,23] and of their parents [25,26,27] could contribute to perpetuating unhealthy lifestyles.

In relation to the actual PA levels of the adolescents who took part in the study, the results showed a high prevalence of sedentarism (82.1%), even more acute among the girls (87.8%), in keeping with what has been reported in literature in the last decade [1,2,3]. Considering the significant number of proven benefits that PA can provide for adolescents’ health [36], these figures represent a considerable threat to public health [37], and their perpetuation proves that the huge effort made by the main institutions and administrations internationally to develop policies promoting PA [38,39] has been, for now, unsuccessful. This underlines the need to continue exploring new research avenues that could serve as a guide and bring about changes in PA habits. 

When we looked at adolescents’ self-perception of PA level, in contrast with the actual levels commented on, it was observed that among both the boys (92.2%) as well as the girls (81.7%), a large majority of the participants considered themselves to be active. The strong tendency to overestimate PA was common to both genders, although higher in the boys, as 90.4% of them and 80.4% of the girls classified as sedentary in accordance with their actual activity level considered themselves to be active. Previous studies had already reported this overestimation phenomenon, both in the child and adult populations, although to a lesser extent, as the percentage of sedentary participants who considered themselves active fell within the range of 38.9–61.2% [40,41,42,43]. In turn, among the few studies considering adolescent populations [14,15], Corder and colleagues found lower percentages of overestimation in the context of Great Britain than those reported here and also observed that this kind of misperception was slightly more prevalent in girls (64.8%) than in boys (60.3%) [14]. To the contrary, in a recent study with 2044 adolescent participants from urban areas of nine different European countries (i.e., Austria, Belgium, France, Germany, Greece, Hungary, Italy, Spain and Sweden), Vanhelst et al. [15] observed a much higher overall overestimation rate (82,9%), very similar to the one obtained in this study (84,9%), while they did not differentiate by gender. Even if, based on this scant existing evidence, it might be premature to try to establish a firm conclusion regarding the influence of gender on the tendency among adolescents to overestimate PA, the results obtained herein would seem to be more in line with previous findings which have constantly recorded lower activity levels and more negative perceptions regarding said activity in girls [44,45].

The high variability observed in overestimation from one study to another might be explained by differences in the selection of participants and/or the methodologies employed to assess actual PA. Further elaborating on these differences, all previous research with adolescents have exhibited large sample sizes, even though the exact number of participants recruited varied greatly from the 799 adolescents in the ROOTS study [14], to both the 2044 in the HELENA study [15] and the 1697 in this study. This different sample sizes had also resulted in a different age coverage between studies, being the ROOTS study [14] centred in early adolescence, while both the HELENA [15] and this studies extended their coverages to late adolescence. By linking the above-mentioned differences in overestimation with the differences in age coverage, it could be hypothesised that overestimation may grow through adolescence. This hypothesis would be consistent with social desirability and social approval bias [46] and also with the important decline of PA during adolescence [47,48]. More importantly still, the geographical areas considered and sampling procedures were not homogeneous either, since: (i) The present study and the one by Corder and colleagues [14] recruited their participants on a regional basis and by intentional non-probabilistic methods, while Vanhelst et al.’s study [15] had a broader European scope and employed random sampling methods; and (ii) the participants in both this and the HELENA studies [15] came from urban areas, while those in the ROOTS study [14] came from urban and rural areas. Relating the differences observed in overestimation with the different environments considered (urban vs rural), it could be stated that overestimation is more prevalent among urban adolescents. This claim is consistent with previous studies which found that adolescents living in rural areas had higher levels of PA [49]. Furthermore, other studies [50,51] have shown a greater amount of moderate to vigorous PA for urban adolescents at the weekends, as a consequence of their participation in organized sports, while lighter activities prevailed among their rural counterparts. This greater intensity would predispose urban adolescents to increased overestimation since it has been suggested that when an individual perceives a bout of PA as intense, tends to report more of it [52]. With regard to the assessment of actual PA, previous studies in adolescents [14,15] used sophisticated automatic monitoring instruments while we used self-reporting procedures. Although accelerometers (or other movement sensors) provide objective and more accurate measures and thus could be preferable, they do not yet fully tackle heterogeneity in the assessments since they incorporate methodological issues related to calibration (cut-off points) and comparability between devices [53,54,55,56].

On the other hand, in light of the results obtained, the perception of parents would also appear to be affected by an overestimation bias, being very similar to that observed for their daughters’ self-perception. This discrepancy between parental perception and the actual PA levels of their children is in keeping with the results shown in previous studies [25,57] and could be a limiting factor when reverting PA values in the young population [26,27]. Furthermore, when analysing parental perceptions in greater depth, it was observed that in both the fathers’ and the mothers’ considerations, cases of overestimation appeared more frequently with respect to the boys and, in turn, cases of underestimation appeared more often with respect to the girls, which suggests that parents apply differentiated perception filters according to the gender of their offspring when assessing their PA level. The analyses carried out in this research have also served to clarify that although a high overestimation of the PA of adolescents can be observed both in their own subjective evaluation and in that of their parents, the level of agreement with regard to father–son/daughter and mother–son/daughter is slight. Therefore, it would be erroneous to assume that perception bias affects them in the same way. On the other hand, the level of agreement in perceptions with regard to father–mother was shown to be good. Considering this information as a whole in a consistent manner with the mediatory role attributed to previous knowledge in the formulation of subjective evaluations [5,6], it can be said that a generational effect could impact the perception of PA levels.

Building strategies to foster an active and healthy lifestyle requires an objective and realistic diagnosis of the baseline situation and; therefore, the discrepancies observed between the adolescents’ actual and perceived PA levels represent a twofold problem. On the one hand, from the point of view of the adolescents, the imbalances in their perception in relation to reality entail a decrease in their susceptibility to change their PA habits, as people need to be aware of behavioural risk factors in order to want to change them and, ultimately, manage to do so [58]. It could also be understood that people who overestimate their PA might disregard PA promotion campaigns, as if these were intended only for inactive people, thus will not respond to them [58]. Further still, the lack of awareness of one’s own achievements or failures when it comes to PA makes it easier for people to adopt an external locus of control, attributing their situation to external forces instead of to their own decisions and abilities, whereby more conformist and apathetic attitudes are established [59]. With regard to the second problem area, the influence that parents might have on their children’s PA levels has been classified in literature as important [28,60,61]. This affirmation is based on the fact that some parents’ abilities, attitudes, behaviours and value judgements [62,63,64], as well as the logistical support they provide [65,66,67], have been confirmed as prominent correlates of children’s PA levels. Consequently, parents who overestimate their children’s PA levels may not provide the support network required to increase their children’s PA levels [26,27,68].

For a proper assessment of the evidence provided by this study, it would appear necessary to describe its limitations. This is a cross-sectional analysis, and the observed associations cannot be interpreted as causal relationships. Although the participants in the study constitute a large sample which represents families from different socio-economic backgrounds, they were chosen using a purposive non-probability procedure and it was not possible to specifically characterize the socio-economic groups that have been covered since socio-economic data were not gathered. Consequently, the risk of biased information being included cannot be completely ruled out. Furthermore, to evaluate the correspondence between actual and perceived PA levels, the former was determined through self-reporting procedures and, although this approach coincides with that most commonly used in literature [27,58], other recent studies have chosen to record PA levels by using accelerometers, with potential implications insofar as the level of correspondence detected is concerned.

## 5. Conclusions

In conclusion, most adolescents and their parents are incapable of correctly assessing the actual PA of the former. More health education campaigns and awareness-raising interventions should be directed at both young people and parents, since it seems that the general population lacks knowledge about the amount of PA that adolescents need to do to achieve a healthy lifestyle.

The gender of the adolescents influences their own assessments of PA and that of their parents, with more negative perceptions regarding PA affecting to the girls. Compared with boys, girls are also more inactive. As a consequence, the modification of PA behaviours among girls is a challenge with particular characteristics that calls for specific interventions.

Future research should further study the PA awareness of adolescents and their mothers and, specifically, longitudinal studies are needed to further study the overestimation phenomenon and contributing factors, and to characterise its evolution across life stages (from childhood to adolescence and into adulthood). Experimental designs are also required to test the efficacy of the different awareness-raising interventions carried out.

## Figures and Tables

**Table 1 ijerph-18-05707-t001:** Sociodemographic characteristics of the parents participating in the study (*N* = 2296).

Characteristic		Father	Mother	Total
Age	*n* valid	1052	1244	2296
	≤35 years	53 (5.0)	86 (6.9)	139 (6.1)
	36–40 years	125 (11.9)	276 (22.2)	401 (17.5)
	41–45 years	330 (31.4)	445 (35.8)	775 (33.8)
	46–50 years	295 (28.0)	323 (26.0)	618 (26.9)
	>50 years	249 (23.7)	114 (9.2)	363 (15.8)
Education level	*n* valid	1040	1231	2271
	Elementary	362 (34.8)	317 (25.8)	679 (29.9)
	Secondary	245 (23.6)	323 (26.2)	568 (25.0)
	Higher	433 (41.7)	591 (48.0)	1024 (45.1)

Note: The values in brackets represent the percentage of valid cases.

**Table 2 ijerph-18-05707-t002:** Description of dependent variables considered.

Variables	Categories
Frequency of PA outside school hours without being clubs or associations	(i) Never; (ii) <1 time/week; (iii) 1 time/week; (iv) 2–3 times/week; (v) 4–6 times/week; (vi) Everyday
PA time outside school hours without being clubs or associations	(i) ½ h; (ii) 1 h; (iii) 2–3 h; (iv) 4–6 h; (v) ≥ 7 h
Frequency of PA outside school in a club or association under supervision	(i) Never; (ii) <1 time/week; (iii) 1 time/week; (iv) 2–3 times/week; (v) 4–6 times/week; (vi) Everyday
PA time outside school in a club or association under supervision	(i) ½ h; (ii) 1 h; (iii) 2–3 h; (iv) 4–6 h; (v) ≥ 7 h
Frequency of PA within the school under supervision	(i) Never; (ii) <1 time/week; (iii) 1 time/week; (iv) 2–3 times/week; (v) 4–6 times/week; (vi) Everyday
Time of PA within the school under supervision	(i) ½ h; (ii) 1 h; (iii) 2–3 h; (iv) 4–6 h; (v) ≥ 7 h
Level of PA according to WHO recommendations	(i) Active; (ii) Sedentary
Self-perception of adolescents’ PA	(i) Not active at all; (ii) Scarcely active; (iii) Quite active; (iv) Active; (v) Very active
Parental perception of adolescents’ PA	(i) Not active at all; (ii) Scarcely active; (iii) Quite active; (iv) Active; (v) Very active

Note. PA = physical activity; WHO = World Health Organization.

**Table 3 ijerph-18-05707-t003:** Characterisation of adolescents’ physical activity (PA).

PA Practice Index		Girls		Boys		χ^2^	*p*
		*N*	%	*N*	%		
Frequency of PA outside school hours without being clubs or associations	Never	66	7.6	58	7.2	112.961	<0.001
	<1 time/week	73	8.6	33	4.1		
	1 time/week	224	26.5	86	10.7		
	2/3 times/week	264	31.2	264	32.9		
	4/6 times/week	120	14.2	182	22.7		
	Everyday	98	11.6	180	22.4		
PA time outside school hours without being clubs or associations	½ h	155	19.7	90	11.9	94.314	<0.001
	1 h	292	37.2	175	23.1		
	2–3 h	244	31.0	288	38.1		
	4–6 h	67	7.7	112	14.8		
	7 h or more	28	3.2	91	12.0		
Frequency of PA outside school in a club or association under supervision	Never	418	49.9	258	32.4	47.311	<0.001
	<1 time/week	15	1.8	10	1.3		
	1 time/week	99	11.8	44	5.5		
	2/3 times/week	156	18.6	237	29.7		
	4/6 times/week	121	14.4	189	23.7		
	Everyday	29	3.3	59	7.4		
PA time outside school in a club or association under supervision	½ h	24	5.6	27	4.9	21.308	<0.001
	1 h	103	23.8	78	14.2		
	2–3 h	181	41.9	228	41.5		
	4–6 h	67	15.5	106	19.3		
	7 h or more	57	6.5	111	20.2		
Frequency of PA within the school under supervision	Never	564	71.4	429	57.4	100.897	<0.001
	<1 time/week	20	2.5	28	3.7		
	1 time/week	35	4.4	77	10.3		
	2/3 times/week	122	15.4	126	16.9		
	4/6 times/week	37	4.7	54	7.2		
	Everyday	12	1.4	33	4.4		
Time of PA within the school under supervision	½ h	58	25.9	77	22.7	14.834	0.005
	1 h	76	33.9	84	24.8		
	2–3 h	70	31.3	114	33.6		
	4–6 h	14	6.3	35	10.3		
	7 h or more	6	2.7	29	8.6		
Level of PA according to WHO recommendations	Active	99	12.2	188	23.7	36.511	<0.001
	Sedentary	714	87.8	604	76.3		

**Table 4 ijerph-18-05707-t004:** Characterisation of perceptions about the level of physical activity of adolescents according to the level of true activity and gender.

Perceived Physical Activity Level		Sedentary		Active	
		Girls	Boys	Girls	Boys
Adolescents	Not active at all	15 (2.1)	4 (0.7)	1 (1.0)	1 (0.5)
	Scarcely active	123 (17.4)	54 (9.0)	8 (8.1)	2 (1.1)
	Quite Active	184 (26.1)	109 (18.3)	14 (14.1)	10 (5.4)
	Active	251 (35.6)	226 (37.9)	40 (40.4)	50 (26.9)
	Very active	132 (18.7)	204 (34.2)	36 (36.4)	123 (66.1)
Mothers	Not active at all	13 (2.4)	7 (1.7)	2 (3.0)	0 (0)
	Scarcely active	98 (17.9)	65 (15.4)	9 (13.6)	14 (10.3)
	Quite Active	126 (23.0)	81 (19.2)	8 (12.1)	14 (10.3)
	Active	199 (36.2)	149 (35.4)	18 (27.3)	54 (39.7)
	Very active	113 (20.6)	119 (28.3)	29 (43.9)	54 (39.7)
Fathers	Not active at all	11 (2.3)	4 (1.1)	1 (2.0)	0 (0)
	Scarcely active	77 (16.4)	53 (14.3)	7 (14.0)	7 (6.3)
	Quite Active	116 (24.7)	80 (21.6)	9 (18.0)	10 (8.9)
	Active	170 (36.2)	120 (32.4)	20 (40.0)	52 (46.4)
	Very active	95 (20.3)	113 (30.5)	13 (26.0)	43 (38.4)

## Data Availability

The data presented in this study are available on request due to privacy restrictions.

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
