# Peer review of "Gender Influences on Physical Activity Awareness of Adolescents and Their Parents"

_ijerph, 2021, doi:10.3390/ijerph18115707_

Round 1

Reviewer 1 Report

General Comment:

The authors should be congratulated for conducting an important study on self-perceived physical activity levels in adolescents.  They highlight a great discrepancy between the evaluation of teenagers' physical activity by themselves or their parents and the desirable physical activity levels according to WHO recommendations. The majority of adolescents and their parents overestimate their current physical activity. In conclusion, the authors hypothesize that this distorted perception may make them less aware of the need to change their physical activity behaviour.  

Specific comments:

Line 48 to 49: missing words

Line 128 to 130: Cronbach's Alpha methodology should be briefly detailed. Only alpha values related to the group of questions used in the present study should be reported

Delete line 131 to 133

Table 2: It is difficult to understand which variables are statistically different between girls and boys. Adding significant p values to this table would also make 3.1 results section easier to read. 

Table 3: kappa values should be added to help reading of section 3.3 

Author Response

Response to Reviewer 1 Comments

R.1. General comment: The authors should be congratulated for conducting an important study on self-perceived physical activity levels in adolescents. They highlight a great discrepancy between the evaluation of teenagers’ physical activity by themselves or their parents and the desirable physical activity levels according to WHO recommendations. The majority of adolescents and their parents overestimate their current physical activity. In conclusion, the authors hypothesize that this distorted perception may make them less aware of the need to change their physical activity behaviour.

Authors’ Response: We appreciate the positive feedback from the reviewer.

R.1. Specific comment 1: Line 48 to 49: missing words.

Authors’ Response: Yes, there was an error. That lines were reworded as follows (now Line 312 to 313): “…people who consider themselves to be more active than they really are have a healthier lifestyle…”

R.1. Specific comment 2: Line 128 to 130: Cronbach’s Alpha methodology should be briefly detailed. Only alpha values related to the group of questions used in the present study should be reported.

Authors’ Response: In order to comply with Reviewer 1 request, lines 810 to 811 were reworded as follows: “…of the instrument, with values of α = 0.89 and α = 0.75 for dimensions 2 and 3, respectively.”

R.1. Specific comment 3: Delete line 131 to 133.

Authors’ Response: That paragraph in Spanish was deleted.

R.1. Specific comment 4: Table 2: It is difficult to understand which variables are statistically different between girls and boys. Adding significant p values to this table would also make 3.1 results section easier to read.

Authors’ Response: Table 2 was renumbered to Table 3 after meeting Reviewer 3 comments and two columns were added to specify c2 and p values.

R.1. Specific comment 5: Table 3: kappa values should be added to help reading of section 3.3.

Authors’ Response: While we would very much like to fulfill Reviewer 1 request, as far as we understand, the addition of kappa values in Table 3 is not feasible for the following reasons:

  • Table 3 provides de descriptive statistics (frequencies and percentages) arising from the cross-tabulation of adolescents’ actual physical activity with adolescents’ and parents’ perceptions with regard to that activity. In this cross-tabulation, actual physical activity is characterized in two categories, sedentary or active, that reflect the degree of compliance with WHO recommendations. However, perceived physical activity was classified in five categories. To provide kappa values in Table 3, the categories used to characterize actual and perceived physical activity needs to be the same and this is only possible if we merge perceived physical activity categories. Doing so would entail a loss of information and we understand that that information is important, as it was used to calculate the odds ratios provided in section 2. Adolescents’ self-perceived levels of physical activity, and especially if it is complemented with the kappa values provided in the text in section 3.3. Correspondence between adolescents’ actual and self-perceived physical activity levels.
  • Kappa values provided in the text were not only calculated as a measure of agreement between actual and perceived physical activity levels, but also were applied as a measure of agreement between adolescents’ and parents’ perceptions and between mothers’ and fathers’ perceptions, too. Furthermore, all of this calculations were conducted overall and separately for boys and girls. Therefore, it would take several tables to present all these kappa values.

As a consequence of this situation, we found preferable to present kappa values in the text and we hope Reviewer 1 can accept this line of argument.

Reviewer 2 Report

need grammar attention

Author Response

Response to Reviewer 2 Comments

R.2. General comment: Need grammar attention.

Authors’ Response: In this new version of the paper, a lot of changes were made for the sake of clarity and concision. Grammar was also checked by a native translator.

R.2. Specific comment 1: Line 53, highlighted text “On the other hand, in order”: pick “on the other hand” or “in order” but no both. I recommend: However, to increase

Authors’ Response: As a result of the extensive changes made in the Introduction to meet Reviewer 3 requirements, this comment no longer applies to the current version of the paper.

R.2. Specific comment 2: Line 54 to 55, highlighted text “underline the importance the perception mothers and fathers might have when it comes to their children abandoning sedentary habits”: need attention... specially grammar

Authors’ Response: The sentence was reworded to (now Line 326 to 328): “highlighted the potentially important role of parents’ perceptions in tackling their children’s sedentary habits”.

R.2. Specific comment 3: Line 58 to 59, highlighted text “], characterized, in children and adolescents,”: punctuation marks.

Authors’ Response: Punctuation marks were incorrectly used so they were removed and the sentence was reworded in a concise manner. Now reads as follows (end of Line 331): “] characterized by including…”.

R.2. Specific comment 4: Line 61, highlighted text “mothers and fathers”: you can use “parents” word as well...

Authors’ Response: “mothers and fathers” was substituted by “parents” (now Line 334).

R.2. Specific comment 5: Line 81, highlighted text “exist”: ???

Authors’ Response: As a result of the extensive changes made in the Introduction to meet Reviewer 3 requirements, this comment no longer applies to the current version of the paper.

R.2. Specific comment 6: Line 91, highlighted text “Galicia”: , Spain?

Authors’ Response: The clarification “(Spain)” was added (now Line 351).

R.2. Specific comment 7: Line 131 to 132: text in Spanish

Authors’ Response: That paragraph in Spanish was deleted.

Reviewer 3 Report

Thank you for the opportunity to review this manuscript. The purpose of the study was to investigate the actual physical activity level of adolescents in comparison to the self-perceived PA level and the perceived PA level by the parents.
I see an acceptable interest of the topic for the readership of the scope of IJERPH. However, the manuscript needs major revisions before considered for potential publication in IJERPH.

Overall

  • The manuscript requires an intensive revision by an institutional native speaker board (e.g., https://www.mdpi.com/authors/english).
  • You tend to write way too long sentences with trying to put too much information in those. Try to help the reader to better follow your work and findings with shorter and clearer writing.

Title

  • I encourage you to reconsider the title. Try to be clearer and more specific.

Abstract

  • The Abstract is written acceptably. However, thorough revision would also be beneficial. Especially, try to be more specific in your results and conclusion part.

Keywords

  • Reconsider your keywords, if they really represent the highlights and key of your study.

Introduction

  • General 1: Try to really stick to the study relevant content in the Introduction.
  • General 2: I miss the clear research gap that you detected, the deduced research questions, and the associated hypotheses of your study.
  • L29-34: This is a rather long section about perception. You can reduce content.
  • L36-42: This is one sentence covering six lines. This is impossible to follow without reading several times.
  • L43-45: I miss an explanation, why this was concluded of the mentioned study results.
  • L45-47: I do not understand, how this sentence is linked to the before unraveled content.
  • L50-51: This is a vague statement without giving us any numbers about the study findings.
  • L53 et seq.: Try to help the reader clearly follow your argumentation towards your research gap, research questions and hypotheses. Try to link the unraveled theoretical content to appearing research gaps, where study in the end of the Introduction necessarily fits in.
  • L73 et seq.: In accordance to the comment before, consider if this very broad global statement should not come earlier in the Introduction.

Methods

  • L92: Can you explain more detailed how you did approach potential participants. Did you randomly pick schools and please tell me what range of education was covered with the different schools? Did you collect any socio-economic data to give better characterization of the socio-economic groups you have covered?
    • Accordingly, did you consider to build sub-groups in terms of age and school the adolescents attended? This would highly enhance your results and findings. You have such a high sample. And I guess there could lie some interesting information in the sub-groups.
  • L103 et seq.: Please revise this part for clarity’s sake; consider bullet points for the various variables; also, in relation to the results options, maybe a Table would also help to appear clearer.
  • L122 et seq.: Do you report any results of the LSQ later on?
  • L131-133: Que pasado aca? La misma pasaje en espanol como antes en ingles? Verdad? Por favor, corigir eso ;-)
  • L135-140: This is written very convoluted. Requires revision.
  • L149-153: I miss a clear description of the proxy variable calculation and in what range it appeared. Of course, I can find the information in that part, however, it is so difficult to read.
  • L157 et seq.: Try to write this section more structured, according to a clear description of the descriptive statistics followed by the inferential statistics.

Results

  • These are some interesting results that you are presenting. Maybe it would enhance the section if you always stay within the similar text structure within a section. Try to not interpret results and write as clear and plain as possible. Thoroughly revise your Results section in accordance.
  • Tables: Try to produce all Tables in same format and editing style.

Discussion and Conclusion

  • The Discussion also needs in-depth and thorough revision. You try to address the most important points of your study with findings in literature, however, your results and apparent discrepancies to other results lack of interpretation and explanation. For example, how do you explain the discrepancies of your results compared to those you refer to in L282-294.
  • Furthermore, try to connect the several section into a bigger picture of your findings towards the end of the Discussion and do not just repeat your findings similar to the Results section.
  • To really being able to follow your argumentation it would also highly benefit if you get the length of your sentences reduced in the Discussion. Even though, sometimes it appears, that by this kind of writing style, you rather blur your thoughts and clear argumentations.
  • This also very obvious in the Conclusion. Herein, the real conclusion, practical implications, and prospective steps should be sketched based on your findings.

Author Response

Response to Reviewer 3 Comments

R.3. General comment: The manuscript requires an intensive revision by an institutional native speaker board (e.g. https://www.mdpi.com/authors/english).

You tend to write way too long sentences with trying to put too much information in those. Try to help the reader to better follow your work and findings with shorter and clearer writing.

Authors’ Response: In this new version of the paper, a lot of changes were made for the sake of clarity and concision. Grammar was also checked by a native translator.

R.3. Specific comment 1: Title: I encourage you to reconsider the title. Try to be clearer and more specific.

Authors’ Response: The title was reconsidered and now reads as follows: “Gender influences on physical activity awareness of adolescents and their parents”.

R.3. Specific comment 2: Abstract: The Abstract is written acceptably. However, thorough revision would also be beneficial. Especially, try to be more specific in your results and conclusion part.

Authors’ Response: The results and conclusion part of the abstract was rewritten as follows (Line 16 to 27): “However, the perceived levels of activity differed significantly from the actual ones with a clear general overestimation both by the adolescents and their parents. When further exploring the data, gender influences were also detected both in adolescent and parental perceptions, since the high rates of overestimation in sedentary individuals were lower in girls and, on the contrary, the low rates of underestimation in active individuals were higher in girls. Moreover, although the level of agreement between actual and perceived physical activity was low overall, with Cohen’s kappa values ranging from 0.006 to 0.047, the lowest values were observed in the case of girls. In conclusion, both the adolescents and their parents were incapable of correctly assessing the actual physical activity of the former, so it seems that the general population lacks knowledge about the amount of physical activity that adolescents need to do to achieve a healthy lifestyle. Consequently, it would be advisable to implement health education campaigns and awareness-raising interventions directed to young people as well as to their parents and, in doing so, gender must be considered by establishing distinct program designs in terms of this variable.”

R.3. Specific comment 3: Keywords: Reconsider your keywords, if they really represent the highlights and key of your study.

Authors’ Response: Keywords were reconsidered and the following were finally selected (Line 28): Gender differences; Physical activity; Assessment imbalances; Adolescents.

R.3. Specific comment 4: Introduction, General 1: Try to really stick to the relevant content in the Introduction.

Authors’ Response: As stated above, all the introduction was rewritten since it was obvious from the reviewer’s comments that the original version was very deficient. We are much more proud of the new version and we appreciate the opportunity of modifying it.

R.3. Specific comment 5: Introduction, General 2: I miss the clear research gap that you detected, the deduced research questions, and the associated hypothesis of your study.

Authors’ Response: Introduction was completely reworked and we hope this version could meet the standards for publication.

R.3. Specific comment 6: Introduction, L29-34: This is a rather long section about perception. You can reduce content.

Authors’ Response: The content about perception was reduced in this new version of the introduction (Line 39 to 41).

R.3. Specific comment 7: Introduction, L36-42: This is one sentence covering six lines. This is impossible to follow without reading several times.

Authors’ Response: Paragraph was streamlined and split in shorter sentences for the sake of clarity and conciseness. Now reads as follows (Line 42 to 46):  “If we talk specifically about perception of PA, subjectivity is also present [8]. That is, when people are asked to assess their own level of PA, their previous education will determine the realism of their assessments. In this way, two people who demonstrate similar PA habits may differ in their assessments in accordance with previously acquired knowledge [9]”

R.3. Specific comment 8: Introduction, L43-45: I miss an explanation, why this was concluded of the mentioned study results.

Authors’ Response: As a consequence of the extensive changes made in the Introduction to solve the problems of the previous version, this comment no longer applies.

R.3. Specific comment 9: Introduction, L45-47: I do not understand, how this sentence is linked to the before unraveled content.

Authors’ Response: “Along these lines” was substituted by “To this regard” for a better linking between sentences (now Line 309).

R.3. Specific comment 10: Introduction, L50-51: This is vague statement without giving us any numbers about study findings.

Authors’ Response: That sentence was deleted since we agree that it was a vague statement.

R.3. Specific comment 11: Introduction, L53 et seq.: Try to help the reader clearly follow your argumentation towards your research gap, research questions and hypotheses. Try to link the unraveled theoretical content to appearing research gaps, where study in the end of the Introduction necessarily fits in.

Authors’ Response: As commented above, the Introduction was streamlined and specific emphasis was placed on highlighting the gaps in previous research that our study intends to fill.

R.3. Specific comment 12: Introduction, L73 et seq.: In accordance to the comment before, consider if this very broad global statement should not come earlier in the Introduction.

Authors’ Response: We agree with the reviewer comment. Consequently, that statement was moved further up to the start of the Introduction (now Line 31 to 34).

R.3. Specific comment 13: Methods, L92: Can you explain more detailed how you did approach potential participants. Did you randomly pick schools and please tell me what range of education was covered with the different schools? Did you collect any socio-economic data to give better characterization of the socio-economic groups you have covered?

Authors’ Response: More details were added in section 2.1. Participants (now Line 352 to 357).

Socio-economic data were not gathered. Consequently, we cannot provide a better characterization of the socio-economic groups that have been covered, and we found appropriate to specifically recognise this deficiency within the limitations to this study (Line 2128 to 2129).

R.3. Specific comment 14: Methods: Accordingly, did you consider to build sub-groups in terms of age and school the adolescents attended? This would highly enhance your results and findings. You have such a high sample. And I guess there could lie some interesting information in the sub-groups.

Authors’ Response: Previous literature suggest that overestimation of physical activity is present not only in adolescents, but also in children and adults [1-5]. Besides, evidence is still scarce in adolescents [6-7] and previous studies, even using large samples as well, have not considered age as a contributing factor, probably because of the high variability in overestimation from one study to another despite considering similar ages (the percentage of inactive participants who perceived themselves as active ranged from 60.3% to 82.9%).

Further to Reviewer 3 suggestion, we decided to conduct a preliminary analysis of our data to explore age effects. To adhere this analysis to the cut-off points of levels of schooling (i.e. first half of junior high school, second half of junior high school, and senior high school), participants were separated in the following age subgroups: i) ≤14 years; ii) 15-16 years; and iii) ≥17 years. As it can be seen below (Table 1), a very high overestimation was observed for the three age groups (with values near the upper end of the range established by previous studies). Furthermore, the variability in overestimation observed here between ages was lower to that on previous literature. Consequently, we do not think that this results deserve a closer look as representative of a real age effect, but they could be attributable to the different number of participants across age subgroups. Finally, we decided not to include this analysis on the final version of the study.

Table 1. Characterization of perceptions about the level of physical activity of adolescents according to the level of true activity and age.

Perceived physical activity level

Sedentary

Active

≤14 years

15-16 years

≥17 years

≤14 years

15-16 years

≥17 years

Sedentary

85

(10.8%)

101

(21.1%)

10

(26.3%)

6

(3.3%)

5

(5.4)

1

(9.1%)

Active

700

(89.2%)

378

(78.9%)

28

(73.7%)

175

(96.7%)

88

(94.6%)

10

(90.9%)

  1. Ronda, G.; Van Assema, P.; Brug, J. Stages of change, psychological factors and awareness of physical activity levels in The Netherlands. Health Promot. Int. 2001, 16, 305-314.
  2. Corder, K.; van Sluijs, E.M.; McMinn, A.M.; Ekelund, U.; Cassidy, A.; Griffin, S.J. Perception versus reality awareness of physical activity levels of British children. J. Prev. Med. 2010, 38, 1-8.
  3. Lechner, L.; Bolman, C.; Van Dijke, M. Factors related to misperception of physical activity in The Netherlands and implications for health promotion programmes. Health Promot. Int. 2006, 21, 104-112.
  4. van Sluijs, E.M.; Griffin, S.J.; van Poppel, M.N. A cross-sectional study of awareness of physical activity: associations with personal, behavioural and psychosocial factors. J. Behav. Nutr. Phys. Act. 2007, 4, 53. doi:10.1186/1479-5868-4-53.
  5. Watkinson, C.; van Sluijs, E.M.; Sutton, S.; Hardeman, W.; Corder, K.; Griffin, S.J. Overestimation of physical activity level is associated with lower BMI: a cross-sectional analysis. J. Behav. Nutr. Phys. Act. 2010, 7, 68. doi:10.1186/1479-5868-7-68.
  6. Corder, K.; van Sluijs, E.M.F.; Goodyer, I.; Ridgway, C.L.; Steele, R.M.; Bamber, D.; Dunn, V.; Griffin, S.J.; Ekelund, U. Physical activity awareness of British adolescents. Pediatr. Adolesc. Med. 2011, 165, 603-609.
  7. Vanhelst, J.; Béghin, L.; Duhamel, A.; De Henauw, S.; Ruiz, J.R.; Kafatos, A.; Manios, Y.; Widhalm, K.; Mauro, B.; Sjöström, M.; Gottrand, F. Physical activity awareness of European adolescents: The HELENA study. Sports Sci. 2018, 36, 558-564.

R.3. Specific comment 15: Methods, L103 et seq.: Please revise this part for clarity’s sake; consider bullet points for the various variables; also, in relation to the results options, maybe a Table would also help to appear clearer.

Authors’ Response: We agree with Reviewer 3 on tabular display of variables would be clear. Consequently, Table 2 was added and the tables below were renumbered accordingly.

R.3. Specific comment 16: Methods, L122 et seq.: Do you report any results of the LSQ later on?

Authors’ Response: Yes, all our results come from the application of the LSQ. Variables described in section 2.2. Variables represent items of LSQ dimensions 2 and 3.

R.3. Specific comment 17: Methods, L131-133: Que pasado aca? La misma pasaje en español como antes en ingles? Verdad? Por favor, corigir eso ;-)

Authors’ Response: That paragraph in Spanish was deleted. We very much appreciate the relaxed tone of your comment.

R.3. Specific comment 18: Methods, L135-140: This is written very convoluted. Requires revision.

Authors’ Response: That paragraph was streamlined (now Line 813 to 820). 

R.3. Specific comment 19: Methods, L149-153: I miss a clear description of the proxy variable calculation and in what range it appeared. Of course, I can find the information in that part, however, it is so difficult to read.

Authors’ Response: That paragraph was reworded to be clear and to include an specific description of the proxy variable calculation (now Line 826 to 837).

R.3. Specific comment 20: Methods, L157 et seq.: Try to write this section more structured, according to a clear description of the descriptive statistics followed by the inferential statistics.

Authors’ Response: That section was restructured as suggested (now Line 838 to 845).

R.3. Specific comment 21: Results: These are some interesting results that you are presenting. Maybe it would enhance the section if you always stay within the similar text structure within a section. Try to not interpret results and write as clear and plain as possible. Thoroughly revise your Results section in accordance.

Authors’ Response: We very much appreciate the interest of Reviewer 3 in our results. A thorough revision of the Results section was carried out. A lot of changes were made for the sake of clarity and concision and to avoid interpretations.

R.3. Specific comment 22: Results, Tables: Try to produce all Tables in same format and editing style.

Authors’ Response: All tables were arranged in the same format and editing style.

R.3. Specific comment 23: Discussion and Conclusion: The Discussion also needs in-depth and thorough revision. You try to address the most important points of your study with findings in literature, however, your results and apparent discrepancies to other results lack of interpretation and explanation. For example, how do you explain the discrepancies of your results compared to those you refer in L282-294.

Authors’ Response: Discussion was thoroughly revised trying to better interpret our results and to explain them in the context of previous literature. As for the example provided in Reviewer 3 comment, a detailed explanation of the discrepancies was added (now Line 1550 to 1875).

R.3. Specific comment 24: Discussion and Conclusion: To really being able to follow your argumentation it would also highly benefit if you get the length of your sentences reduced in the Discussion. Even though, sometimes it appears, that by this kind of writing style, you rather blur your thoughts and clear argumentations.

Authors’ Response: Writing was revised not only in the Discussion but all along the paper. A lot of effort was placed in reducing the length of sentences and on setting up a clear argumentation. We hope this new version will meet your approval.

R.3. Specific comment 25: Discussion and Conclusion: This also very obvious in the Conclusion. Herein, the real conclusion, practical implications, and prospective steps should be sketched based on your findings.

Authors’ Response: Conclusions were reworked as suggested (no Line 2137 to 2151).

Round 2

Reviewer 2 Report

I have no further comments

Author Response

Dear reviewer, 

Thank you again for your comments on our manuscript finally entitled “Gender influences on physical activity awareness of adolescents and their parents” (Manuscript ID: ijerph-1198523). 

We have examined this minor revision comments and have made corresponding corrections that we hope will meet with your approval. Please see below, in blue, our detailed response to comments. 

Kind regards,

María A  Fernández-Villarino

Reviewer 3 Report

Thank you for the in-depth and thorough work you put into the revised version of the manuscript. I completely see that you addressed most of the comments in your manuscript that led to an improvement that makes the manuscript valid for publication.

However, I have a few minor comments that need to be addressed before accepting the manuscript for publication.

Introduction:

General: The quality of your Introduction highly improved. Yet I have some lingering comments for better understanding of the readership. In the revised version, you form some very long sentences including various statements. It is understandable, however, your Introduction and the readership would benefit if you reduce the amount of such long sentences by short statements with reduced content.

L32-34: Revise sentence for better understanding or form two sentences.

L39-42: Reduce length of sentence or write clearer.

L42-44: This statement requires referencing.

L62-65: This statement is pretty long and very hard to get. Please revise into two statements.

L70-74: Way too long. Please revise into two statements.

Methods

I acknowledge the work the authors put into the Methods section.

Thanks for Table 2, that is pretty meticulous work for a better overview of your dependent variables you have collected.

The Methods section is fine!

Results

Results section’s sub-titles: Please avoid abbreviations in titles throughout your manuscript.

Apart the meticulous work put in the Results section is acknowledged and includes all necessary content and provides the expected quality required for a potential publication.

Discussion and Conclusion

Good work, some minor lingering comments:

L275-277: Sentences’ value is hard to get.

L279-287: What is your deeper contribution to your findings, when listing the various sample sizes of different studies? You do not provide an explanation of that fact. I do not see a contribution to the general Discussion by saying: there are other studies having different sample sizes from different areas. So what? What does this contribute to your study or how can you separate your results/findings from theirs?

L332 et seq.: It is rather unusual telling us the strength of your study here. I think you have told us before-hand anyways that you are study has a lot of strength, haven’t you? Or would you consider a scientific publication of your study without being convinced that your study and results have scientific soundness?

Conclusions

Good revision of the original Conclusion section.

Author Response

Dear reviewer,

Thank you again for your comments on our manuscript finally entitled “Gender influences on physical activity awareness of adolescents and their parents” (Manuscript ID: ijerph-1198523). The fact that the reviewers seem to be satisfied with our efforts in addressing previous comments was very encouraging for us and we appreciate a lot this new opportunity to further improve our paper.

We have examined this minor revision comments and have made corresponding corrections that we hope will meet with your approval. Please see below, in blue, our detailed response to comments. All line numbers refers to the manuscript file with the tracked changes.

Kind regards,

María A. Fernández-Villarino

Response to Reviewer 3 Comments

R.3. General comment: Introduction: The quality of your Introduction highly improved. Yet I have some lingering comments for better understanding of the readership. In the revised version, you form some very long sentences including various statements. It is understandable, however, your Introduction and the readership would benefit if you reduce the amount of such long sentences by short statements with reduced content.

Authors’ Response: In this new version of the Introduction, several changes were made to fulfill reviewer’s recommendations. Specific details about these changes can be found below in our responses to R.3 Specific comments 1-5.

R.3. Specific comment 1: L32-34: Revise sentence for better understanding or form two sentences.

Authors’ Response: That sentence was revised and now reads as follows: “Subsequent studies describe only a slight evolution in the data [2]. Considering gender, the situation is particularly bad for girls, who increased their prevalence of physical inactivity, even setting it at over 83% [3].” (L32-34).

R.3. Specific comment 2: L39-42: Reduce length of sentence or write clearer.

Authors’ Response: That sentence was shortened as follows: “As the basic concept of this body of research, perception refers to sensory awareness and depends on the knowledge and opinions held by the individual about what it is being assessed [5-7].” (L39-41)

R.3. Specific comment 3: L42-44: This statement requires referencing.

Authors’ Response: The statement was downplayed to stay closer to Lechner and colleagues [9], who were also referenced (L:42-44). Renumbering subsequent references was also required.

  1. Lechner, L.; Bolman, C.; Van Dijke, M. Factors related to misperception of physical activity in The Netherlands and implications for health promotion programmes. Health Prom. Int. 2006, 21, 104-112.

R.3. Specific comment 4: L62-65: This statement is pretty long and very hard to get. Please revise into two statements.

Authors’ Response: The statement was revised into two statements. Now reads as follows: “Only a few studies have analysed the PA awareness of European adolescents [14,15]. According to their findings, 60.3% to 80.9% of adolescents believe that they are physically active when they are not.” (L62-65)

R.3. Specific comment 5: L70-74: Way too long. Please revise into two statements.

Authors’ Response: As recommended, that passage was revised into two statements. Now reads as follows: “Notwithstanding, previous studies have scarcely considered potential gender influences over the realism of adolescents’ assessments. To this regard, Corder and colleages [14] found that, compared with girls, boys were less likely to overestimate their PA.” (L69-72)

R.3. Specific comment 6: Methods: I acknowledge the work the authors put into the Methods section.

Authors’ Response: The authors would like to thank the reinforcement provided by the reviewer.

R.3. Specific comment 7: Thanks for Table 2, that is pretty meticulous work for a better overview of your dependent variables you have collected.

Authors’ Response: We also would like to thank Reviewer 3 for advising a table as a better way to provide an overview of the variables considered in our study. We feel this yielded a great improvement.

R.3. Specific comment 8: The Methods section is fine!

Authors’ Response: Thank you again for your encouraging comment.

R.3. Specific comment 9: Results section’s sub-titles: Please avoid abbreviations in titles throughout your manuscript.

Authors’ Response: Titles were revised to avoid abbreviations.

R.3. Specific comment 10: Apart the meticulous work put in the Results section is acknowledged and includes all necessary content and provides the expected quality required for a potential publication.

Authors’ Response: This positive feedback is very important for us. Thank you for taking the time not only to identify the downsides but also to acknowledge the upsides of our paper.

R.3. Specific comment 11: L275-277: Sentences’ value is hard to get.

Authors’ Response: The sentence was reworded to: “The high variability observed in overestimation from one study to another might be explained by differences in the selection of participants and/or the methodologies employed to assess actual PA.” (L355-357)

R.3. Specific comment 12: L279-287: What is your deeper contribution to your findings, when listing the various sample sizes of different studies? You do not provide an explanation of that fact. I do not see a contribution to the general Discussion by saying: there are other studies having different sample sizes from different areas. So what? What does this contribute to your study or how can you separate your results/findings from theirs?

Authors’ Response: We greatly appreciate the opportunity to further explain the implications of our results. The following two passages were added for this purpose:

  • L360-367: This different sample sizes had also resulted in a different age coverage between studies, being the ROOTS study [14] centered in early adolescence while both the HELENA [15] and this studies extended their coverages to late adolescence. By linking the above-mentioned differences in overestimation with the differences in age coverage, it could be hypothesised that overestimation may grow through adolescence. This hypothesis would be consistent with social desirability and social approval bias [46] and also with the important decline of PA during adolescence [47,48].
  • L373-382: Relating the differences observed in overestimation with the different environments considered (urban vs rural), it could be stated that overestimation is more prevalent among urban adolescents. This claim is consistent with previous studies which found that adolescents living in rural areas had higher levels of PA [49]. Furthermore, other studies [50,51] have shown a greater amount of moderate to vigorous PA for urban adolescents at the weekends, as a consequence of their participation in organized sports, while lighter activities prevailed among their rural counterparts. This greater intensity would predispose urban adolescents to increased overestimation since it has been suggested that when an individual perceives a bout of PA as intense, tends to report more of it [52].

As a consequence, it was also required to add the following references (and also renumbering subsequent ones):

46.  Hebert, J.R.; Ma, Y.; Clemow, L.; Ockene, I.S.; Saperia, G.; Stanek, E.J.; Merriam, P.A.; Ockene, J.K. Gender differences in social desirability and social approval bias in dietary self-report. Am. J. Epidemiol. 1997, 146, 1046- 55.

47.  Corder, K.; Sharp, S.J.; Atkin, A.J.; Griffin, S.J.; Jones, A.P.; Ekelund, U.; van Sluijs, E.M. Change in objectively measured physical activity during the transition to adolescence. Br. J. Sports Med. 2015, 49, 730-6.

48.  Ortega, F.B.; Konstabel, K.; Pasquali, E.; Ruiz, J.R.; Hurtig-Wennlöf, A.; Mäestu, J.; Löf, M.; Harro, J.; Bellocco, R.; Labayen, I.; Veidebaum, T.; Sjöström, M. Objectively measured physical activity and sedentary time during childhood, adolescence and young adulthood: a cohort study. PLoS One 2013, 23, 8e60871.

49.  Regis, M.F.; Oliveira, L.M.F.T.D.; Santos, A.R.M.D.; Leonidio, A.D.C.R.; Diniz, P.R.B.; Freitas, C.M.S.M.D. Urban versus rural lifestyle in adolescents: associations between environment, physical activity levels and sedentary behavior. Einstein (São Paulo) 2016, 14, 461-467.

50.  Machado-Rodrigues, A.M.; Coelho-E-Silva, M.J.; Mota, J.; Padez, C.; Martins, R.A.; Cumming, S.P.; Malina, R.M Urban–rural contrasts in fitness, physical activity, and sedentary behaviour in adolescents. Health Promot. Int. 2014, 29, 118-129.

51.  Donatiello, E.; Russo, M.D.; Formisano, A.; Lauria, F.; Nappo, A.; Reineke, A.; Sparano, S.; Barba, G.; Russo, P.; Siani, A. Physical activity, adiposity and urbanization level in children: results for the Italian cohort of the IDEFICS study. Public Health 2013, 127, 761-765.

52.  Welk, G.J.; Kim, Y.; Stanfill, B.; Osthus, D.A.; Calabro, A.M.; Nusser, S.M.; Carriquiry, A. Validity of 24-h physical activity recall: physical activity measurement survey. Med. Sci. Sports Exer. 2014, 46, 2014-2024.

R.3. Specific comment 13: L332 et seq.: It is rather unusual telling us the strength of your study here. I think you have told us before-hand anyways that you are study has a lot of strength, haven’t you? Or would you consider a scientific publication of your study without being convinced that your study and results have scientific soundness?

Authors’ Response: That passage containing the strength of our study was removed (L456-457).

R.3. Specific comment 14: Conclusions: Good revision of the original Conclusion section.

Authors’ Response: We would like to thank once more your interest in our study and your supportive comments.
